# DNA-Templated Fluorescent Silver Nanoclusters Inhibit Bacterial Growth While Being Non-Toxic to Mammalian Cells

**DOI:** 10.3390/molecules26134045

**Published:** 2021-07-01

**Authors:** Lewis Rolband, Liam Yourston, Morgan Chandler, Damian Beasock, Leyla Danai, Seraphim Kozlov, Nolan Marshall, Oleg Shevchenko, Alexey V. Krasnoslobodtsev, Kirill A. Afonin

**Affiliations:** 1Nanoscale Science Program, Department of Chemistry, University of North Carolina at Charlotte, Charlotte, NC 28223, USA; lrolband@uncc.edu (L.R.); mchand11@uncc.edu (M.C.); dbeasock@uncc.edu (D.B.); ldanaino@uncc.edu (L.D.); sskozlov04@gmail.com (S.K.); oshevche@uncc.edu (O.S.); 2Department of Physics, University of Nebraska at Omaha, Omaha, NE 68182, USA; lyourston@unomaha.edu (L.Y.); nmarshall@unomaha.edu (N.M.)

**Keywords:** DNA, silver, AgNC, nanocluster, antibacterial, fluorescence

## Abstract

Silver has a long history of antibacterial effectiveness. The combination of atomically precise metal nanoclusters with the field of nucleic acid nanotechnology has given rise to DNA-templated silver nanoclusters (DNA-AgNCs) which can be engineered with reproducible and unique fluorescent properties and antibacterial activity. Furthermore, cytosine-rich single-stranded DNA oligonucleotides designed to fold into hairpin structures improve the stability of AgNCs and additionally modulate their antibacterial properties and the quality of observed fluorescent signals. In this work, we characterize the sequence-specific fluorescence and composition of four representative DNA-AgNCs, compare their corresponding antibacterial effectiveness at different pH, and assess cytotoxicity to several mammalian cell lines.

## 1. Introduction

The formation of silver nanoclusters (AgNCs) on single-stranded (ss) DNA templates has been shown to promote the unique optical properties defined by the sequences of the DNA strands [1,2,3]. Out of all available coordination sites on nucleobases, silver cations demonstrate the highest affinity for the N3 of cytosines, and therefore cytosine-rich ssDNAs become efficient capping-agents for AgNC formation [4,5,6]. The size and shape of AgNCs are regulated by rationally designed and chemically synthesized short DNA oligonucleotides with different numbers of single-stranded cytosines embedded in secondary and tertiary DNA structures such as hairpin loops, i-motifs, and G-quadruplexes, to name a few [4,5,6]. The optical properties of DNA-AgNCs are dictated by their size, as the appearance of the characteristic fluorescence is possible for nanoclusters comprised of only a few silver atoms. At this nanometer size, a continuous density of electronic energy states present in bulk silver breaks up and a band gap in the material becomes apparent [1,2,3,4,5,6,7,8]. This, in turn, causes a molecule-like behavior of AgNCs with discrete energy states allowing for size-dependent fluorescence to occur [7,8,9]. DNA-capped AgNCs are also generally more resistant to photobleaching when compared to traditional organic fluorophores or fluorescent proteins, and this property begets the application of DNA-AgNCs in a variety of nanophotonics and biosensing/biomedical applications [10,11,12,13]. While nanophotonics and biosensing with DNA-AgNCs’ advantageous optical properties have been widely probed and studied, other practical uses of AgNCs remain unexplored. Since the main functional component of DNA-AgNCs is silver, applications based on effects known for this element may prove useful. Various forms of silver, including ions and silver nanoparticles, are well-documented to have antibacterial efficacy [14,15,16,17]. Several groups have shown DNA-AgNCs to be effective against both Gram-negative and Gram-positive bacteria in liquid cultures [18,19,20] and against the formation of biofilms when aptamers for increased targeting and binding of the bacteria of interest were introduced [21,22]. However, the underlying mechanisms and relationship between the fluorescent properties of DNA-AgNCs and their antibacterial actions are still understudied and poorly understood. AgNCs offer a large surface-to-volume ratio and are composed of both forms of silver: cationic (Ag^+^) and neutral (Ag^0^), thereby providing further advantages over solid silver, silver salts, or silver nanoparticles.

We reason that understanding and linking optical and antibacterial properties of DNA-AgNCs may pave the way to the development of next generation antibacterial agents with high potency and regulated activity. Our current work includes four representative DNA hairpins that template the formation of DNA-AgNCs with four distinct colors and investigates their optical properties in relation to antibacterial activity measured at different pH, as well as in relation to cytotoxicity assessed for several human cell lines. The use of antibacterial DNA-AgNCs formed on DNA hairpins becomes advantageous for various antibacterial formulations and opens broader possibilities for DNA nanotechnology due to the relative structural stability of the hairpins and their inability to participate in any undesirable base-pairings, thus not interfering with any other DNA nanodesigns. As proof-of-concept work, we explore the use of DNA-AgNC forming hairpins with odd numbers of consecutive cytosines (C7, C9, C11, or C13) in their loop compositions.

## 2. Results

### 2.1. Template Design

Cytosine-rich ssDNAs are suitable capping agents for templating stable DNA-AgNCs due to cytosine’s high affinity for silver ions, Ag^+^. Various sequences have been reported to stabilize clusters with unique optical properties, including bright emission bands in the visible part of the spectrum and excitation bands in the UV and visible regions. The emission wavelengths can be modulated by choosing a specific DNA sequence and various colors (e.g., yellow, green, orange, red as shown in Figure 1) of DNA-AgNCs have been reported based on prevalent emission wavelengths for a particular nanocluster.

While the parameters that define emissive properties of DNA-AgNCs are still not well understood, it is generally accepted that shape, size, and overall charge state of the AgNC are among the main contributors. We hypothesize that the same factors are responsible for modulating the antibacterial activity of the DNA-AgNCs. While cytosine-rich ssDNAs have been widely used in synthesizing AgNCs, these sequences are prone to forming alternative DNA structures [23]. Such alternative structures include i-motif and non-canonical C-Ag-C base paring facilitated by the presence of silver cations. We have compared properties of two C12-containing templates in which one is an opened ssDNA and another is a sequence embedded in a hairpin loop. The results (Appendix A) clearly show the differences observed for these two sequences after DNA-AgNC formation. As evident from AFM images (Appendix A), the single-stranded template shows various degrees of polymerization, while the hairpin-loop template does not polymerize, forming individual DNA-AgNCs. These results agree well with our recent study demonstrating that the formation of alternative DNA structures in the presence of Ag^+^ drives the polymerization of various sequences containing single-stranded (ss) C-rich stretches [23]. Additionally, such polymerization also alters fluorescence properties of AgNCs (Appendix A). Hairpin-looped structures feature one single fluorescence peak for C_12_ sequence while single-stranded templates show multiple peaks, suggesting the formation of AgNCs with various sizes and shapes due to variety of the polymerized templates [23]. To avoid structural and functional uncertainties associated with ssC_N_ template sequences and to make the structures suitable for further implementation in nanodesign, we have chosen to work only with DNA hairpin templates wherein the C_N_ sequence forms the loop of the hairpin structure. Four representative templates were constructed with the same double-stranded stem and a loop with a variable number of ssCs (C7, C9, C11, and C13). This design gradually increases the number of binding sites for silver and makes the size of the loop larger (Figure 1) [3,24]. An odd number of cytosines in the loop with +2C steps was intended to noticeably alter properties of the DNA-AgNCs with fluorescent colors covering the entire visible spectral region (Figure 1). We reasoned that such substantial optical differences would provide an insight into which factors modulate the antibacterial activity of the DNA-AgNCs and how antibacterial activity correlates with the optical signatures of individual DNA-AgNCs. Incubation of the looped DNA templates with silver nitrate and subsequent reduction of silver using sodium borohydride results in the formation of optically active nanoclusters with bright emissions (Figure 1 and Figure 2). Appendix A shows UV-Vis spectra of all four purified AgNC samples immediately after their formation.

### 2.2. Fluorescence

The formation of DNA-AgNCs is tracked by the changes in solution that are observed after the addition of silver nitrate and sodium borohydride reducing agent followed by incubation in the dark for 24 h. We have characterized the optical properties of these DNA-AgNCs using fluorescence excitation-emission matrix spectroscopy (EEM). EEM represents the excitation–emission relationships of the optical response of the DNA-AgNCs presented as 2D contour maps [25]. Figure 2 shows EEM maps for all four DNA-AgNCs in the 300–800 nm range for the excitation while recording the emission spectrum spanning 300–800 nm wavelengths. Initial readings of the EEMs after purification (top panel of Figure 2) indicates that all four samples show a great degree of similarity in the behavior of emission. All four samples are dominated by one peak in the red region of the spectrum. While similar in general, the peaks show detectable differences. 

In Table 1, we summarize peak positions for the maximum excitation and maximum emission for all four DNA-AgNCs. It appears that smaller loop DNA-AgNCs have maxima for both excitation and emission shifted to longer wavelengths. It is very pronounced for C7 and C9 with λ_EXC_/λ_EM_ = 600/685 nm and λ_EXC_/λ_EM_ = 580/661 nm, respectively. Further shifts to λ_EXC_/λ_EM_ = 562/647 nm and λ_EXC_/λ_EM_ = 562/645 nm are observed for C11 and C13, respectively. C11 DNA-AgNC also features an extra shoulder of emission at shorter excitation wavelengths. C9, C11, and C13 peaks appear to be elongated featuring red edge emission shift (REES) as reported previously and is common for AgNCs [23,26,27]. Interestingly, C7 DNA-AgNC does not have REES-based elongation of the emission peak that is well-pronounced for other samples. These observations emphasize the differences of DNA-AgNCs formed by the four looped templates despite all samples having “red” emission. 

The differences in observed optical properties intensify further as samples are allowed to age. Changes in emission pattern with time develop very quickly during “maturation” stage of the AgNCs typical for our preparation procedure. These changes represent the conversion of AgNCs from “red” to “green” emission upon interaction with ambient conditions as we have previously reported [26]. ”Red” to “green” conversion can be linked to the interactions of AgNCs with species dissolved in the solution that are capable of oxidizing silver atoms (Ag^0^ → Ag^+^), such as molecular oxygen [26]. Many reports documented the “blue” shift with aging and some protocols call for bubbling oxygen through to stimulate this transition [28]. All four of our samples also experience such “blue” shift and eventually develop a pattern of multi-peaked emission spectra with some samples more noticeable than others (for example, C9 vs. C13). Figure 2 summarizes in detail all changes in emission patterns when C7-C13 DNA-AgNCs age over a period of two weeks. The appearance of additional emission peaks in the “green” region are obvious for C7, C9, and C11, while C13 remained primarily as a single peak. The changes in C13 DNA-AgNCs include the loss of elongated shape with the near-IR part of the peak disappearing over time. Shorter loops C7, C9, and C11 develop an obvious multipeak excitation–emission pattern over time. These new peaks appear in the “orange” and “green” spectral regions. Additionally, these peaks differ significantly in their position and intensity. C7 DNA-AgNCs have only one new peak of λ_EXC_/λ_EM_ = 465/547 nm—“green.” The intensity of this new peak is 27% the intensity of the original “red” peak. Both C9 and C11 have multi-peak patterns of newly appeared “orange” and “green” emission labeled O (longer wavelengths) and G (shorter wavelengths). The positions of these two new peaks are very similar for both C9 and C11 DNA-AgNCs: λ_EXC_/λ_EM_ = 475/606 nm (O-C9), λ_EXC_/λ_EM_ = 480/606 nm (O-C11), λ_EXC_/λ_EM_ = 408/530 nm (G-C9), and λ_EXC_/λ_EM_ = 410/536 nm (G-C11). Similar spectral positions indicate the same nature of “green” states for both C9 and C11 DNA-AgNCs. The differences for these two samples, however, include the position of the “red” peak as listed in Table 1. Another major difference is the relative intensities of orange and green peaks, O and G. O peak dominates in the C11 sample, while G is more pronounced in the C9 sample. The following are the relative intensities as compared to the original “red” peak: 22% (O-C9), 360% (O-C11), 112% (G-C9), and 29% (G-C11). The observed intensities suggest that C9 DNA-AgNCs primarily stabilize G state while C11 DNA-AgNCs prefer O. The O peak is not observed in the fluorescence of the aged C7 sample, while C13 remains “red” during aging.

To complete the description of the emissive properties of C7-C13 DNA-AgNCs, we also visualized the emission of nanoclusters under UV excitation on a trans-illuminator (at 254 nm). Such excitation is typically discussed as a means of excitation via DNA bases that contact silver atoms in the nanocluster. The 254 nm excitation results in a color palette of the employed samples (colors under trans-illuminator excitation, Figure 1—top). This picture reflects the rich emission pattern observed for C7-C13 DNA-AgNCs in the visible part of the spectrum. 

### 2.3. Cell Culture Experiments

To assess the relative effects of representative DNA-AgNCs on bacterial cells, TOP10F’ *E. coli* are grown in liquid cultures and treated with the panel of DNA-AgNCs at a final concentration of 4 µM DNA (Figure 3A). A decrease in the bacterial growth is observed over a 20 h period for all *E. coli* cultures treated with DNA-AgNCs when compared to the non-treated control. There is a strong dose-dependence noted for all DNA-AgNCs (Appendix A) with 4 μM to be the lowest DNA concentration that shows reasonable antibacterial effectiveness for all constructs. As such, all experiments are carried out at 4 μM in order to best resolve differences between four tested DNA-AgNCs. To quantitatively examine the inhibition of *E. coli* growth, we compare the changes in the amount of time (Δt_1/2_) required for bacteria cultured with each DNA-AgNC to grow to half of their maximum optical density when referenced to untreated cells from the same experimental group. C13 is the most effective DNA-AgNC at slowing bacterial growth as the calculated Δt_1/2_ for C7, C9, C11, and C13, shown with their 95% confidence intervals, are 5.5 ± 0.2 h, 5.8 ± 0.3 h, 8.8 ± 0.2 h, and 10.7 ± 0.2 h, respectively. The effect of free silver at the concentrations used to synthesize the DNA-AgNCs can be considered minimal since the control experiment with 650 μM of Ag^+^, the highest of the concentrations used for C_N_ synthesis, has a minimal effect on the growth curve after being reduced with NaBH_4_ (Appendix A). The antibacterial effect of the DNA-AgNCs greatly increases at lower pH. When *E. coli* grown in pH 5.5-buffered LB are treated with 4 μM of each DNA-AgNC, the growth is nearly fully inhibited over the entire 20 h (Figure 3A). From these experiments, the performance of C7 and C9 as antibacterial agents are similar, though it is clear that C13 outperforms C11 significantly. From these data, it appears that there may be a correlation between the number of cytosines and the antibacterial efficacy of the DNA-AgNC for larger hairpins.

To test the same conditions in mammalian cells, we use several human cell lines and all DNA-AgNCs are again introduced at a 4 µM final concentration. The cell viability is assessed after 20 h of incubation using an MTS assay (Figure 3B). No statistically significant reduction in cell viability is observed after incubation with DNA-AgNCs for Jurkat, THP1, or 293FT cells. To ensure the safety of DNA-AgNCs for mammalian cells, the same experiments are repeated at a final concentration of 8 µM DNA-AgNC with all three cell lines and the results are similar (Appendix A). Therefore, AgNCs remain non-toxic to mammalian cells at concentrations two-fold higher than required to efficiently inhibit bacterial growth.

### 2.4. Stoichiometry Determination

In order to quantify the number of silver atoms bound to each ssDNA oligonucleotide, energy dispersive X-ray spectroscopy (EDS) elemental analysis is performed and micrographs of dried AgNC solutions are recorded using scanning electron microscopy. The ratio of the relative atomic percentages of the Ag and P calculated from the EDS spectrum are used for evaluating the stoichiometric ratio of silver per hairpin-loop DNA template. From these experiments, we determine that each C7 DNA-AgNC binds an average of 9.9 ± 0.6 silver atoms (± SEM), C9 DNA-AgNC binds an average of 8.5 ± 0.5 silver atoms, C11 DNA-AgNC binds an average of 11.7 ± 0.5 silver atoms, and C13 DNA-AgNC binds an average of 10.2 ± 0.9 silver atoms. The differences in the number of silver atoms bound were not found to be statistically significant between each templating strand. A representative micrograph of a DNA-AgNC sample is shown in Figure 4, and the micrographs of all samples analyzed by EDS are shown in the Appendix A.

## 3. Discussion

Silver has long been used as a disinfectant. The most recent applications include the use of silver nanoparticles in many different areas including food packaging, water and air disinfection, the textile sector, and medical applications (Silver Soaker^®^ Catheters, Acticoat™, SilvaSorb^®^ Gel) [29,30]. The search for new therapeutic agents to combat multi-drug resistant bacteria is ongoing. While silver nanoparticles have been recently extensively studied for their use as antibacterial agents [31], novel silver nanoclusters have been largely overlooked [18] primarily because most studies have focused on biosensing applications due to the unique optical properties of AgNCs [27,32,33]. DNA-AgNCs have several advantages which position them as excellent candidates for antibacterial applications. First, DNA-AgNCs are small in size—they are comprised of only a few atoms of silver capped with stabilizing cytosine-rich ssDNA oligonucleotides. Since DNA-AgNCs are bound to DNA, in addition to serving as a host for AgNCs, DNA can also be utilized for embedding AgNCs into a structured network of functional assemblies leading to novel properties and functions of hybrid nanomaterials [23,26]. In this study, we show that AgNCs are capable of inhibiting bacterial growth at a much lower concentration (4 µM) than carbenicillin (132 µM), which is a bactericidal antibiotic from the penicillin group and was used as a positive control in this study. We confirm, herein, that DNA-AgNCs show very little toxicity against human cells. In addition to serving as a template for AgNC synthesis, DNA templates may further contribute to better solubility and biocompatibility of clusters. Thus, DNA-AgNCs could potentially be used against a broad range of various bacteria without harmful side effects. Furthermore, the robust fluorescence of AgNCs can be coupled with biocompatibility and antibacterial properties to produce label-free bioimaging agents with dual purpose. 

Our additional experiments indicate that DNA-AgNCs exhibit antibacterial activity against *Lactobacilli* (Appendix A), suggesting that DNA-AgNCs might cause undesirable effects to the gut microbiota, and should be avoided in applications involving oral ingestion of these materials, e.g., in food packaging. This is especially concerning, given the increased antibacterial efficacy of AgNCs at lower pH (Figure 3A). Previous work has shown that decreasing the pH will increase the antibacterial efficacy of silver nanoparticles that likely arises from an increased production of reactive oxygen species and that DNA-AgNCs are capable of catalyzing the production of reactive oxygen species [18,34]. It is possible, therefore, that DNA-AgNCs produce an increased amount of reactive oxygen species as pH decreases. Another possibility for the increased antibacterial activity of DNA-AgNCs at lower pH is leaching silver into solution through cytosine protonation. However, the pK_a_ of cytosine is close to 4.4, so less than 10% of cytosines would be expected to be protonated at pH = 5.5 [35]. Additionally, DNA-AgNCs retain their fluorescence pattern over a wide pH range, down to a pH of 5, implying stabile character of the DNA-AgNCs at pH = 5.5 [36]. Therefore, we consider it unlikely that the decreased pH would cause silver ions to leach from the AgNC causing enhanced inhibition of bacterial growth.

We show that C13 DNA-AgNC produces the highest antibacterial activity among the four studied DNA template sequences followed by C11, while C7 and C9 DNA-AgNCs show lower activity (Figure 3A). While several factors might contribute to antibacterial activity, the number of silver atoms comprising the AgNCs does not appear to be the decisive factor. Using EDS elemental analysis, we confirmed that there is no statistical significance in the number of silver atoms bound by each DNA-AgNC (Figure 4A). It is unlikely that the amount of silver in the DNA-AgNCs determines their antibacterial properties as all templates stabilize clusters of nearly the same size with *N*≈10-11 bound in them. Our observation is supported by a previous report which also ruled out the amount of silver atoms per cluster [18]. Emission color has been proposed to correlate with DNA-AgNCs’ antibacterial properties with “red” emissive clusters being the most active [18]. We also turned to fluorescence properties in search of a possible explanation for the antibacterial activity of DNA-AgNCs. It is unclear how exactly DNA-AgNCs act in terms of antibacterial properties and this uncertainty in their mechanism of antibacterial action has triggered the current study. There are clear changes in fluorescent properties of DNA-AgNCs which we can correlate with the increased antibacterial activity of AgNCs. It appears that a single-peak emission pattern might be the key. The mere presence of “red” fluorescence does not define antibacterial properties; all samples are “red” initially, but the abilities to inhibit bacterial growth differ among C13, C11 and C9, C7. C13 remains “red” during aging while C11 effectively converts to “orange.” At the same time, C13 provides better antibacterial efficacy as compared to C11. Additionally, C7 remains primarily “red” while its activity is lower than C11 and C13. C9 is the only sample that develops a “green” peak with high intensity, but it is also less effective at inhibiting bacterial growth. As such, we hypothesize that the stability of the “red” fluorescence upon aging may be one of the most decisive factors for the antibacterial efficacy of DNA-AgNCs.

It is commonly accepted that DNA-AgNCs include both Ag^0^ and Ag^+^ atoms in their composition. The ratio of Ag^0^/Ag^+^ defines the overall charge state and the color of the nanocluster’s emission [37]. It has been proposed that distinct “green” and “red” fluorescence occurs for a “magic number” of neutral silver atoms in the nanocluster [37,38]. Four neutral atoms produce green fluorescence and six Ag^0^ atoms produce red fluorescence regardless of the number of Ag^+^ [37]. Recent studies indicated that such conversion does not change the overall number of silver, *N*, in the cluster as this conversion is reversible [26,31]. 

“Red,” “orange,” and “green” emissive states of AgNCs that we observe may represent different ratios of Ag^+^ to Ag^0^. Aging of the samples can therefore be explained by the interaction of AgNCs with species dissolved in the solution that are capable of oxidizing silver atoms (Ag^0^ → Ag^+^). For example, dissolved “molecular oxygen” might effectively convert “red” to “orange” and to “green” emitting species. Controlled oxidation with hydrogen peroxide confirms our conclusion (Appendix A). The addition of hydrogen peroxide gradually converts emissive patterns which resembles “aging” of all samples. We have previously confirmed that gradual aging or oxidation due to addition of hydrogen peroxide can be reversed by re-reduction of resultant AgNCs [26]. This process is reversible and can be done multiple times suggesting that red-ox state of the AgNCs rather than size plays a critical role in fluorescence pattern that we observe here for hairpin-loop templated DNA-AgNCs. Many studies relate the antibacterial activity of silver nanoparticles to the oxidative release of Ag^+^ [39]. In this regard, DNA-AgNCs already have silver ions in their composition and can therefore act as antibacterial agents. Furthermore, the ratio of Ag^+^/Ag^0^ can modulate the antibacterial activity of DNA-AgNCs. It is tempting to suggest that the increased number of silver ions in DNA-AgNCs may explain higher antibacterial activity. However, C9 is dominated by “green” emitting species which would supposedly have the highest number of cations in the DNA-AgNC composition, while we observe that C9 has the second lowest antibacterial effect. It is possible that the looped hairpin templates used herein while varying in length may have different protective properties for DNA-AgNCs depending on the final conformation of the loop wrapping around silver nanoclusters. For example, faster conversion of C13 to a non-emitting species might indicate lesser protection of the clusters and thus correlate better with higher antibacterial activity of C13. Additional studies will be required to identify whether intact DNA-AgNCs act as the antibacterial agent or if their activity requires nanocluster dissolution with the release of silver ions into the solution.

Another possible explanation for different activities observed for the four analyzed samples is that the nature of emissive and non-emissive states may play a role. We observed that all four samples age and react with hydrogen peroxide very differently (Appendix A). It is also apparent that partial oxidation is involved in “red” to “orange” or “red” to “green” conversion of emissive DNA-AgNC states. We have evaluated the rate of “red” peak conversion as a function of hydrogen peroxide concentration (Figure 5) for all four DNA-AgNCs using a modified Stern–Volmer relationship (Equation (1)) [40].
(1)F0F=(1+KDCH2O2)(1+KSCH2O2)

Generally, a linear Stern–Volmer plot indicates a single class of fluorophores which are all equally vulnerable to quenching by hydrogen peroxide [40]. All four DNA-AgNC samples show non-linear F_0_/F vs. H_2_O_2_ concentration dependence (Figure 5). This suggests a complex nature of fluorescence quenching with at least two deactivation pathways: intermolecular quenching due to H_2_O_2_ (dynamic quenching, K_D_, most likely due to intersystem crossing from singlet S* to triplet T*) and intramolecular conversion of “red” to “orange” or “green” (static quenching, K_S_, due to change of the overall charge state of AgNC). While C7 and C9 DNA-AgNCs exhibit slightly upward curvature, C11 and C13 show clear downward curvature. This observation indicates that these two groups have different mechanisms of quenching while interacting with hydrogen peroxide. Typically, downward curvature is associated with fluorophores which are inaccessible to the quencher, suggesting a more protective nature of larger loops [40]. Interestingly, C13 also shows the highest antibacterial activity and the largest downward curvature among the studied C-loop templates. While it is difficult to specify exact details of interactions between the “quencher” and certain states of AgNCs without further studies, it is apparent that the charge state of the DNA-AgNC can play a critical role in defining the antibacterial activity of nanoclusters. Several reports have indicated that silver nanoclusters are capable of generating excessive amounts of intracellular reactive oxygen species, which is proposed as the major contributing factor defining DNA-AgNCs’ antibacterial ability [18,41]. It is also becoming increasingly apparent from recent studies both theoretical [42] and experimental [43] that certain shape, composition, and charge states of DNA-AgNCs can increase the chances of optically “dark” states to exist with highspin multiplicity (doublet and triplet). Therefore, it is not unreasonable to propose that highspin AgNC states may interact with highly abundant triplet oxygen removing the “spin-forbidden” condition and stimulating the transition of triplet to singlet oxygen: ^3^O_2_ → ^1^O_2_. Since singlet oxygen is far more reactive compared to triplet oxygen, this can explain various antibacterial properties of different DNA-AgNCs and the generation of reactive oxygen species. 

In conclusion, nucleic acid-based nanomaterials are often designed based on two rationales: the delivery of functional moieties that can be implemented into the nanoscaffolds and the patterns in recognition of nucleic acids which contribute to the cellular response. DNA-AgNCs offer an approach by which functional fluorescent moieties can contribute to selective growth inhibition of bacterial cultures. The results of this study suggest that the rich optical behavior of the DNA-AgNCs may be tightly linked to the antibacterial properties of this novel class of nanostructures. Excitation–emission pattern, interconversion of emissive states, and their connection with environmental changes are the keys to understanding the mechanism of DNA-AgNC inhibitive action. The results obtained herein warrant further exploration of the antibacterial effects of DNA-AgNCs on both pathogenic and non-pathogenic bacteria species. 

## 4. Materials and Methods

*Synthesis of DNA-AgNCs.* All DNA oligonucleotides were purchased from Integrated DNA Technologies (IDT), Inc. (Coralville, IA, USA) as desalted products and used without further purification. All sequences are listed in the Appendix A. Nuclease-free water was obtained from IDT. Sodium borohydride was purchased from TCI America, Inc. (Portland, OR, USA). In a typical preparation, DNA template (C13, C11, C9, or C7) and AgNO_3_ aqueous solutions were mixed and incubated for 25 min at room temperature in ammonium acetate buffer (100 mM NH_4_OAc, pH 6.9). Next, NaBH_4_ aqueous solution was added and samples were placed on ice and stirred vigorously. The final concentrations (C) of the components were as follows: C_DNA-template_ = 50 μM; C_AgNO3_ was adjusted to match the number of cytosines in the loop according to n*AgNO_3_:C_n_;C_NaBH4_:C_AgNO3_ was taken at 1:1 ratio and C_NH4Ac_ = 4 mM. The solution was then incubated in the dark for 24 h at 4 °C. Synthesized DNA-AgNCs were then purified via a NAP-5 (Cytiva) filtration gel column purchased from Sigma-Aldrich, Inc. (Saint Louis, MO, USA) for fluorescence measurements. Purification was performed according to the protocol supplied by the manufacturer. Final concentrations of DNA-AgNCs obtained after filtration varied between 8–15 μM and were evaluated by taking DNA absorption at 265 nm wavelength. For antibacterial and mammalian cell viability experiments, DNA-AgNCs were purified using 3 kDa Amicon centrifugal filters by washing twice with buffer and diluting to 50 µM. 

*Fluorescence measurements.* The excitation and emission spectra were acquired on a Duetta–Fluorescence and Absorbance Spectrometer (Horiba, Inc., Chicago, IL, USA). In all the measurements, the concentration of the templating sequence was kept the same at ~6 µM. Fluorescence measurements were carried out in a Sub-Micro Fluorometer Cell, model 16.40F-Q-10 (from StarnaCells, Inc., Atascadero, CA, USA) at room temperature of ~22 °C. The excitation–emission matrix spectra (EEMS) were recorded with 0.5 nm resolution. Fluorescence spectra were recorded with the emission wavelength range from 300 nm to 1000 nm, the initial excitation wavelength was set to 280 nm, and the final excitation wavelength was set to 800 nm with an increment of 0.5 nm. Matrix data were then used for 2D contour plot using MagicPlot Pro software (v2.9, Magicplot Systems, LLC, Saint Petersburg, Russia).

*Bacterial growth assays.* TOP10F’ *E. coli* were purchased from ThermoFisher Scientific (Walham, MA, USA) and grown in Luria broth (LB) purchased from Sigma. Where shown, the pH of LB was adjusted to pH 5.5 with 100 mM 2-morpholin-4-yl ethanesulfonic acid (MES). *E. coli* were grown in LB from single colonies while shaking at 200 rpm at 37 °C in a GeneMate Incubated Shaker (VWR International, LLC, Radnor, PA, USA). For treatment with AgNCs, bacteria were diluted in LB to an optical density at 600 nm (OD_600_) of 0.018–0.020. Next, 50 μL of diluted bacteria were added to each well of a 96-well flat-bottom, black-walled plate. Purified DNA-AgNCs were added with LB to reach a final volume of 100 μL in each well with 4 μM final concentration of DNA-AgNCs. Carbenicillin was used as a positive control at a final concentration of 50 µg/mL (132 μM). The lids of the plates were hydrophobically treated by filling them with 10 mL of 20% ethanol, 0.05% Triton X-100 for 30 s [44]. The excess liquid was drained, and the lid was leaned against the back of a fume hood to dry for 30 min [44]. The lids were parafilmed to the microwell plates to prevent excess evaporation. Microplate optical density measurements were recorded using a Tecan Spark (Tecan Group Ltd., Männedorf, Zürich, Switzerland) microwell plate reader. The plates were shaken for 30 s between each measurement and were incubated at 37 °C with OD_600_ measurements taken every 15 min over 20 h. A minimum of six technical repeats and three biological repeats of each experiment were performed. The time required for each growth curve to reach half its maximum optical density, t_1/2_, was calculated with GraphPad Prism 9 (San Diego, CA, USA) using a non-linear fit of the data. The difference between the untreated control t_1/2_ and the treatment t_1/2_ is reported as Δt_1/2_. Additional experiments in *Lactobacillus* cultures were conducted to understand the effects of DNA-AgNCs on normal microflora (Appendix A).

*Mammalian cell viability assays*. For all experiments, cells were maintained and cultured at 37 °C, 5% CO_2._ THP1-Dual™ cells were purchased from InvivoGen (San Diego, CA, USA) and were maintained in RPMI 1640, 2 mM l-glutamine, 25 mM HEPES, 10% heat-inactivated fetal bovine serum (FBS), and PenStrep (100 U/mL,100 μg/mL). 293FT cells were cultured in DMEM, 2 mM l-glutamine, 10% FBS, and PenStrep (100 U/mL, 100 µg/mL). Jurkat cells were cultured in RPMI 1640, 2 mM l-glutamine, 25 mM HEPES, 10% FBS, and PenStrep (100 U/mL, 100 μg/mL). For cell viability studies, cells were plated in a 96-well flat-bottom plate at a density of 40,000 cells per well along with DNA-AgNC solution at final concentrations of 4 or 8 μM and final well volumes of 100 μL. After incubation with AgNC treatments for 20 h, 20 µL of CellTiter 96^®^ AQueous One Solution Cell Proliferation Assay (MTS) were added to each well. Plates were incubated for an additional 75 min at 37 °C, 5% CO_2_. The plates were then read on a Tecan Spark microplate reader for absorbance at 490 nm. Sixteen reads per well were averaged for each value.

*Scanning electron microscopy (SEM) and energy-dispersive X-ray spectroscopy (EDS) elemental analysis.* Solutions of 50 µM C7, C9, C11, and C13 in buffer were pipetted onto a polished silicon wafer as 10 µL droplets. Droplets were allowed to dry in a covered Petri dish overnight at room temperature. The solid residue was analyzed with SEM/EDS to determine the atomic ratio between P and Ag. Dried solutions on the Si substrate were analyzed with a JEOL JSM-6480 SEM. Micrographs were taken in secondary electron mode with an accelerating voltage of 5 kV. EDS spectra were collected using an Oxford Instruments INCA EDS behind a beryllium window. Atomic percentages were calculated by the INCA instrument software based on the intensities of phosphorus K𝛼 (2.013 keV) and silver L𝛼 (2.984 keV) characteristic X-rays.

*Statistical analysis.* All data is presented as the mean ± standard deviation or standard error of the mean (specified for each case) for a minimum of *N* = 3 independent biological replicates. For statistical analysis, a one-way ANOVA was performed, followed by a *t*-test using GraphPad Prism 9.0.0 Software for Windows. *P*-values of *p* < 0.05 were considered statistically significant.

## Figures and Tables

**Figure 1 molecules-26-04045-f001:**
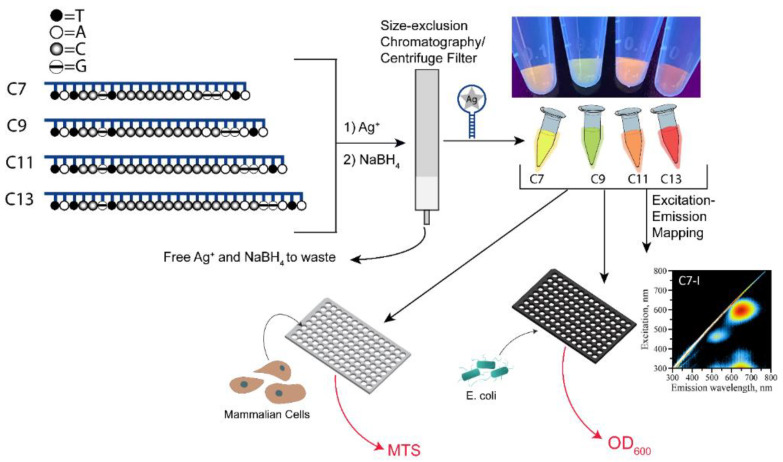
Experimental flow of DNA-AgNC synthesis, purification, and analysis. The embedded image shows DNA-AgNCs after their purification upon UV excitation on a transilluminator.

**Figure 2 molecules-26-04045-f002:**
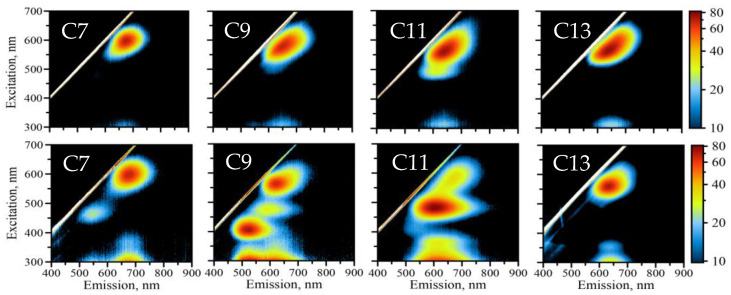
Fluorescence measurements of DNA-AgNCs. Excitation–emission matrix spectroscopy: top panel shows the initial readings of freshly made DNA-AgNCs, bottom panel corresponds to the analysis of samples aged over a period of two weeks (dual fluorescence pattern with both green and red peaks is typical for freshly prepared AgNCs producing distinct colors shown in Figure 1).

**Figure 3 molecules-26-04045-f003:**
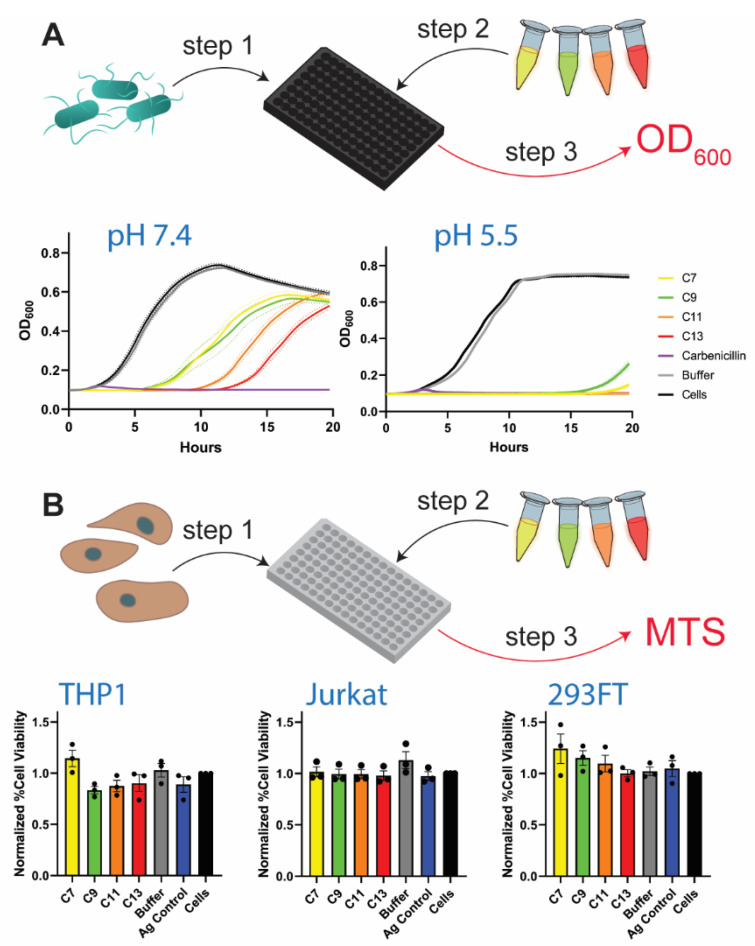
Biological activity of DNA-AgNCs. (**A**) The growth curves of *E. coli* when treated with 4 μM DNA-AgNC are shown at pH 7.4 (the pH of standard LB) and pH 5.5. The standard error of the mean of each measurement is shown as a dotted line on both sides of the solid line in the same color. The lines for C11, C13, and Carbenicillin overlap at pH 5.5. (**B**) The normalized cell viability of THP1-Dual^TM^, Jurkat, and 293FT cells after incubation with 4 μM AgNC for 20 h, as assessed by MTS assay.

**Figure 4 molecules-26-04045-f004:**
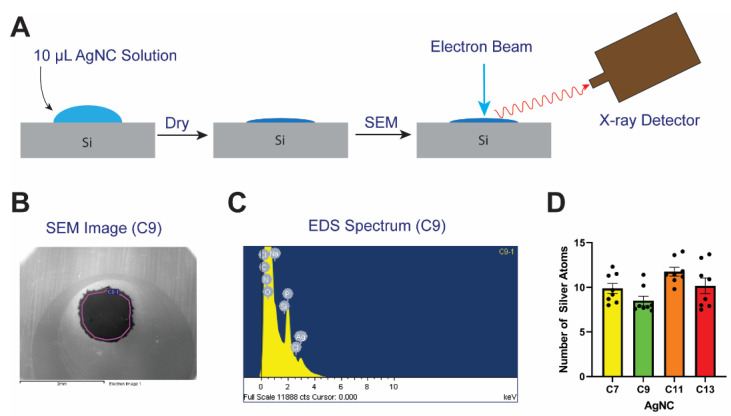
(**A**) The general workflow for the SEM and EDS experiments is shown. The drying is done at ambient conditions with the silicon wafer covered by a Petri dish. (**B**) A representative SEM image is shown of a DNA-AgNC C9 sample and (**C**) the raw EDS spectrum of the same sample. The purple line in (**B**) defines the outer perimeter of the area that is scanned to obtain the EDS spectrum. (**D**) The number of bound silver atoms on each templating DNA hairpin is shown as determined by EDS. Error bars shown as a dotted line on both sides of the solid line are the standard error of the mean for each measurement.

**Figure 5 molecules-26-04045-f005:**
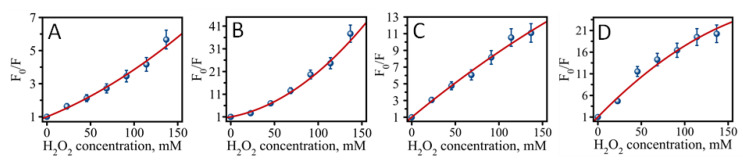
Stern-Volmer plots for (**A**) C7 DNA-AgNCs, (**B**) C9 DNA-AgNCs, (**C**) C11 DNA-AgNCs, (**D**) C13 DNA-AgNCs. Data points were fitted with Equation (1), which considers two possible quenching mechanisms: static and dynamic.

**Table 1 molecules-26-04045-t001:** Spectral position of excitation and emission for “red” emitting peak in initial EEMs.

Wavelength, nm	C7	C9	C11	C13
**EXC_MAX_**	600	580	562	562
**FLU_MAX_**	685	661	647	645

## Data Availability

All data are available and can be shared upon request.

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
