# Peer review of "DNA-Templated Fluorescent Silver Nanoclusters Inhibit Bacterial Growth While Being Non-Toxic to Mammalian Cells"

_molecules, 2021, doi:10.3390/molecules26134045_

Round 1

Reviewer 1 Report

This study is aimed to further our understanding on optical properties of small DNA templated silver nanoclusters, their applicability as antibacterial agents and possible correlations between the optical and antibiotic properties. For this purpose, four different DNA hairpins containing a 7 base pairs stem and a 7-13 loop of silver(I) ion binding 2´-deoxycytidines. The advantage of haipin structure has been shown to be that the silver nanoclusters do not tend to polymerize as readily as those formed on linear templates.

Among the nanocusters prepared by siver(I) ion binding followed by sodium borohydride reduction, the cluster containing 13 cytosine bases and around 10 silver atoms appeared to be most efficient antibiotic agent, but the other clusters studied were not markedly less efficient. The clusters were not toxic in mammalian cell lines, but inhibited growth of Lactobacilli, suggesting undesirable effects to gut microbes.

Some correlations between the optical properties and bilogical activity seems to exist.

The paper is well written. The presentation is logic and rather detailed. The experiments are described in sufficient detail and the conlusions appear justified. The results still are somewhat tentative but novel, and they undoubtedly are a step forward in this interesting field.

I have only two suggestions:

I think that the labels for the top row pictures number 3 and 4 from the left should be C11 and C13, not C1.

Secondly, in the list of references, usually full names of the journals are given, but not always. Sometimes abbreviations are used. Please, be systematic.

Author Response

We are thankful for the reviewer’s comments and have made the two requested adjustments to our manuscript.

Reviewer 2 Report

The authors propose a synthesis of silver nanoclusters templated with different DNA strands, and study their fluorescence properties and antibacterial activity.

The paper is well written and the data are interestings. 

I think that the manuscript should be improved in some aspects:

  • data on the characterization of the nanoclusters are missing: no UV-Vis spectra are present, and TEM imaging is absent as well. What's the size of the nanoclusters? Their distribution and polidispersity? 
  • it is not clear if this kind of nanomaterials have some advantage when compared to more traditional AgNP (citrate coated, PVP coated, etc.) Is there an advantage related to the presence of DNA?
  • Why is antibacterial effect higher at lower ph? Is this pH dangerous for bacterial cells? Usually antibacterial effect are evaluated at physiological pH, other values have, in my opinion, a much limited relevance
  • how is the number of silver atoms per DNA hairpin determined? is that simpy based on a C/Ag ratio? Having no clue on the size of nanoclusters, and therefore on the total number of silver atoms, it is not clear how this value was determined
  • the authors state that "DNA contributes to the solubility and biocompatibility" How? no data on solubility are reported, and a comparison on biocompatibility with/without DNA should be made
  • I think the discussion of the paper should be completely reworked: the authors claim a correlation between the fluorescent properties of the nanoclusters and their antibacterial activity. Even if the data on both are good and well-presented, no correlation could be inferred between the two. None of the experiments demonstrates that fluorescence is related to the antibacterial acitivity, so I think that the two properties should be presented independently, and any relation between them should be presented as an unsupported hypotesis.

I don't think the manuscript can be accepted in its present form, I suggest to do a major revision.

Author Response

The authors propose a synthesis of silver nanoclusters templated with different DNA strands, and study their fluorescence properties and antibacterial activity.

The paper is well written and the data are interesting. 

I think that the manuscript should be improved in some aspects:

  • data on the characterization of the nanoclusters are missing: no UV-Vis spectra are present, and TEM imaging is absent as well. What's the size of the nanoclusters? Their distribution and polydispersity? 

The size of AgNC, being only 10-11 atoms, falls below spatial resolution of TEM. While being a method of choice for characterization of larger silver nanoparticles, TEM is not common in characterizing AgNCs comprised only of several atoms. We did, however, per reviewer’s request include UV-Vis spectra and several excitation/emission spectra for all the samples as Supporting Figure S8 in the Supporting Information.

  • it is not clear if this kind of nanomaterials have some advantage when compared to more traditional AgNP (citrate coated, PVP coated, etc.) Is there an advantage related to the presence of DNA?

We believe that silver nanoclusters (AgNCs) as novel nanomaterials might have some advantages over rather well characterized silver nanoparticles (AgNP) when it comes to antibacterial activity. DNA-templated silver nanoclusters reported herein are only 10-11 atoms in size and have larger surface to volume ratio as well larger Ag+ to Ag0 ratio. The direct comparison of AgNC and AgNP is rather challenging at this stage since there are still many unknowns for AgNCs such as size, shape, and charge state. We thank the reviewer for this suggestion and will certainly plan to compare AgNC/AgNP once we have more information about AgNCs and their properties.   

  • Why is antibacterial effect higher at lower ph? Is this pH dangerous for bacterial cells? Usually antibacterial effect are evaluated at physiological pH, other values have, in my opinion, a much limited relevance

We appreciate reviewer’s comment regarding limited relevance of “non-physiological” pH values. Our motivation for testing lower pH stems from the following main reason: some bacteria modify their environment during growth to locally lower pH which could be used to selectively trigger antibacterial activity of AgNC due to the effect we observed in this study. We have now added this argument in the discussion section of the paper to make it clearer as per reviewer’s suggestion.

  • how is the number of silver atoms per DNA hairpin determined? is that simpy based on a C/Ag ratio? Having no clue on the size of nanoclusters, and therefore on the total number of silver atoms, it is not clear how this value was determined

The number of silver atoms per each DNA hairpin was determined using the ratio of intensities between P (phosphorus) peak and Ag (silver) peak in EDS spectra of TEM analysis. These data are described and presented in Figure 4, Figure S5, Table S1.

  • the authors state that "DNA contributes to the solubility and biocompatibility" How? no data on solubility are reported, and a comparison on biocompatibility with/without DNA should be made

The reviewer is correct – we did not report solubility and biocompatibility comparative data with and without DNA. Unfortunately, silver nanoclusters do not exist without a template which in our case is DNA. Therefore, it would not be possible to make such a comparison. Our statement is suggestive rather than definite; thus, we rewrote this statement which now reads as “In addition to serving as a template for AgNC synthesis, DNA might further contribute to better solubility and biocompatibility”.    

  • I think the discussion of the paper should be completely reworked: the authors claim a correlation between the fluorescent properties of the nanoclusters and their antibacterial activity. Even if the data on both are good and well-presented, no correlation could be inferred between the two. None of the experiments demonstrates that fluorescence is related to the antibacterial acitivity, so I think that the two properties should be presented independently, and any relation between them should be presented as an unsupported hypotesis.

Our discussion section was structured to provide several possible hypotheses to the observed experimental data. It is quite possible, per reviewer’s opinion, that fluorescent properties and antibacterial properties may not be correlated, however, it is also possible that they may be correlated. This correlation of fluorescent with antibacterial properties has been proposed before and we wished to discuss this in the context of previously published studies. We do acknowledge, however, that some arguments in the discussion have been stated rather than hypothesized. We have now corrected these statements to reflect speculative nature of the arguments.

Reviewer 3 Report

The authors present a highly interesting study on the use of DNA-templated silver nanoclusters (AgNCs) as antibacterial agents. DNA-templated AgNCs are known for their reproducible fluorescence, depending on the identity of the DNA oligonucleotide. In the present study, the authors correlate luminescence properties with antibacterial activity. By using four closely related oligonucleotide sequences, DNA-templated AgNCs of different “color” were synthesized and evaluated. Even though all conjugates contain roughly the same number of silvers, they display different fluorescence and different antibacterial activity. The authors discuss in detail different possible explanations for this. Fortunately, the conjugates are non-toxic to human cell lines, suggesting possible future applications as antibacterial agents.

The study was performed at the highest scientific level. All conclusions are fully substantiated. Several reference measurements are included in the supporting information. I gladly recommend acceptance of this manuscript.

Minor points:

The authors may want to include references to the two experimental structures of DNA-templated AgNCs (JACS 2019, 141, 11465 and Angew. Chem. Int. Ed. 2019, 58, 17153).

The fact that different fluorescence patterns are observed after two weeks likely correlates to the presence of different DNA-templated AgNCs. Did the authors try to separate and characterize these, e.g. by HLPC and mass spectrometry?

Do the authors have any idea why the AgNCs are more potent antibacterial agents at lower pH? Are the cytosine residues being protonated, so that they release silver ions?

Author Response

The authors present a highly interesting study on the use of DNA-templated silver nanoclusters (AgNCs) as antibacterial agents. DNA-templated AgNCs are known for their reproducible fluorescence, depending on the identity of the DNA oligonucleotide. In the present study, the authors correlate luminescence properties with antibacterial activity. By using four closely related oligonucleotide sequences, DNA-templated AgNCs of different “color” were synthesized and evaluated. Even though all conjugates contain roughly the same number of silvers, they display different fluorescence and different antibacterial activity. The authors discuss in detail different possible explanations for this. Fortunately, the conjugates are non-toxic to human cell lines, suggesting possible future applications as antibacterial agents.

The study was performed at the highest scientific level. All conclusions are fully substantiated. Several reference measurements are included in the supporting information. I gladly recommend acceptance of this manuscript.

Thank you for your support.

Minor points:

The authors may want to include references to the two experimental structures of DNA-templated AgNCs (JACS 2019, 141, 11465 and Angew. Chem. Int. Ed. 2019, 58, 17153).

Thank you for suggesting these papers to us. We have now included these as references.

The fact that different fluorescence patterns are observed after two weeks likely correlates to the presence of different DNA-templated AgNCs. Did the authors try to separate and characterize these, e.g. by HLPC and mass spectrometry?

This is an interesting suggestion. We did not attempt to separate and characterize these as we believe that the different fluorescence patterns that are observed after 2 weeks are a result of the AgNCs becoming oxidized. We have shown that titrating H2O2 into the DNA-AgNC solution produces similar fluorescence changes as the two-week wait (Supporting Figure S7). Additionally, in previous work (DOI: 10.1039/d0nr03589k) we have shown that DNA-AgNCs can be re-reduced and re-oxidized changing color patterns reversibly.1 This similarity in changes of fluorescence pattern and its reversibility suggest critical involvement of red-ox state of the DNA-AgNCs rather than their size. We also confirmed that a similar size of AgNCs is formed on all four studied here templates. We acknowledge, however, that a more comprehensive study associated with AgNC size, AgNC charge, and their relation to antibacterial properties might be very valuable for the community. We plan to initiate such a study in the near future.   

Do the authors have any idea why the AgNCs are more potent antibacterial agents at lower pH? Are the cytosine residues being protonated, so that they release silver ions?

This is a great question. We are not entirely certain as to why DNA-AgNCs may inhibit bacterial growth more at lower pH. The pKa of the most acidic site of the cytosine has been experimentally determined to be 4.4, so less than 10% of nucleobase will be protonated at this pH=5.5 (DOI: 10.1021/jp8068877).2 Additionally, the fluorescent properties of other DNA-AgNCs have been shown to be stable across the range of pH 5-9, implying that the silver is unlikely to be released or oxidized by the change in pH alone (DOI: 10.1088/2050-6120/ab47f2).3 It is known that fluorescent DNA-templated nanoparticles are able to produce reactive oxygen species (DOI: 10.1021/acsami.6b00670).4 Previous reports have also shown that the antibacterial efficacy of silver nanoparticles, as well as their production of reactive oxygen species, increases as pH decreases (DOI: 10.1021/acsami.7b17274).5 While our future work may address the production of reactive oxygen species by DNA-AgNCs as a function of pH, it is outside of the scope of the current manuscript.

  1. Yourston, L.; Rolband, L.;  West, C.;  Lushnikov, A.;  Afonin, K. A.; Krasnoslobodtsev, A. V., Tuning properties of silver nanoclusters with RNA nanoring assemblies. Nanoscale 2020, 12 (30), 16189-16200.
  2. Verdolino, V.; Cammi, R.;  Munk, B. H.; Schlegel, H. B., Calculation of pKa Values of Nucleobases and the Guanine Oxidation Products Guanidinohydantoin and Spiroiminodihydantoin using Density Functional Theory and a Polarizable Continuum Model. The Journal of Physical Chemistry B 2008, 112 (51), 16860-16873.
  3. Gambucci, M.; Cerretani, C.;  Latterini, L.; Vosch, T., The effect of pH and ionic strength on the fluorescence properties of a red emissive DNA-stabilized silver nanocluster. Methods and Applications in Fluorescence 2019, 8 (1), 014005.
  4. Javani, S.; Lorca, R.;  Latorre, A.;  Flors, C.;  Cortajarena, A. L.; Somoza, Á., Antibacterial Activity of DNA-Stabilized Silver Nanoclusters Tuned by Oligonucleotide Sequence. ACS Applied Materials & Interfaces 2016, 8 (16), 10147-10154.
  5. Tian, X.; Jiang, X.;  Welch, C.;  Croley, T. R.;  Wong, T.-Y.;  Chen, C.;  Fan, S.;  Chong, Y.;  Li, R.;  Ge, C.;  Chen, C.; Yin, J.-J., Bactericidal Effects of Silver Nanoparticles on Lactobacilli and the Underlying Mechanism. ACS Applied Materials & Interfaces 2018, 10 (10), 8443-8450.

Round 2

Reviewer 2 Report

The authors addressed my concerns only in a marginal extent. I still think that the characterization on the size of the nanoclusters and the DNA/silver ratio is unclear, and the discussion has not been changed significantly.

I therefore can't change my opinion on the manuscript.

Author Response

N/A